# The Forest Line Mapper: A Semi-Automated Tool for Mapping Linear Disturbances in Forests

**Gustavo Lopes Queiroz** , **Gregory J. McDermid \*** , **Mir Mustafizur Rahman** and **Julia Linke**

Applied Geospatial Research Group, Geography Department, University of Calgary,
Calgary, AB T2N 4V8, Canada; glopesqu@ucalgary.ca (G.L.Q.); mmrahm@ucalgary.ca (M.M.R.);
julia.linke@ucalgary.ca (J.L.)
\* Correspondence: mcdermid@ucalgary.ca; Tel.: +1-403-220-4780

**Abstract:** Forest land-use planning and restoration requires effective tools for mapping and attributing linear disturbances such as roads, trails, and asset corridors over large areas. Most existing linear-feature databases are generated by heads-up digitizing. While suitable for cartographic purposes, these datasets often lack the fine spatial details and multiple attributes required for more demanding analytical applications. To address this need, we developed the Forest Line Mapper (FLM), a semi-automated software tool for mapping and attributing linear features using LiDAR-derived canopy height models. Accuracy assessments conducted in the boreal forest of Alberta, Canada showed that the FLM reliably predicts both the center line (polyline) and footprint (extent polygons) of a variety of linear-feature types including roads, pipelines, seismic lines, and power lines. Our analysis showed that FLM outputs were consistently more accurate than publicly available datasets produced by human photo-interpreters, and that the tool can be reliably deployed across large application areas. In addition to accurately delineating linear features, the FLM generates a variety of spatial attributes associated with line geometry and vegetation characteristics from input canopy height data. Our statistical evaluation indicates that spatial attributes generated by the FLM may be useful for studying and classifying linear features based on disturbance type and ground conditions. The FLM is open-source and freely available and is aimed to assist researchers and land managers working in forested environments everywhere.

**Keywords:** LiDAR; mapping; forestry; vegetation; corridor; disturbance; software; open source

## 1. Introduction

Roads, trails, and asset corridors permeate forest ecosystems around the world, providing access to settlements, resources, and recreation areas through densely vegetated terrain. In recent decades, the volume of linear disturbances has proliferated, driven by growing human populations and mounting demand for resources. The effects of these disturbances on natural ecosystems are varied and complex [1–3]; managing them requires geospatial tools that can map and attribute linear disturbances quickly and effectively over large areas.

In the forested regions of Alberta, Canada, the exploration and recovery of natural resources has created an extensive network of linear disturbances. Roads, railways, transmission lines, pipelines, and seismic lines (petroleum-exploration corridors) facilitate the access and extraction of wood, minerals, and petroleum products throughout the province. The most abundant of these disturbances are seismic lines; linear-access trails cut through forests to allow the placement of geophones and other equipment associated with subsurface petroleum exploration [4]. Depending on the equipment used to clear them, seismic lines range in width from ~1.5 m to 10 m and can occur at densities exceeding 40 km/km$^2$ [4,5].

Seismic lines and other linear disturbances in Alberta exert cumulative environmental effects on vegetation communities [6–11], microclimate [12,13], wildlife [14–16], hydrology [17–19], and carbon dynamics [20,21]. In response, leading resource companies are identifying lines that could benefit from restoration treatments designed to restrict access and encourage a return to forest cover [20]. However, linear disturbances are difficult to map using automated techniques, and most existing databases are constructed manually through heads-up digitizing (i.e., tracing a cursor over digital images). Inspected carefully, these delineations often contain spatial generalizations that may limit their effectiveness for detailed restoration planning (Figure 1A). For example, most linear features are represented in geodatabases as centerlines (polylines) that do not accurately represent the physical extent of disturbances on the ground. Buffering these centerlines to create polygons—a common GIS practice among land managers and researchers—often creates further spatial uncertainties. In addition to these spatial limitations, most linear-feature databases contain a limited number of attributes—if any—meaning (for example) that narrow disturbances with heavy regrowth (Figure 1C) are often undifferentiated from wide disturbances with less vegetation (Figure 1B).

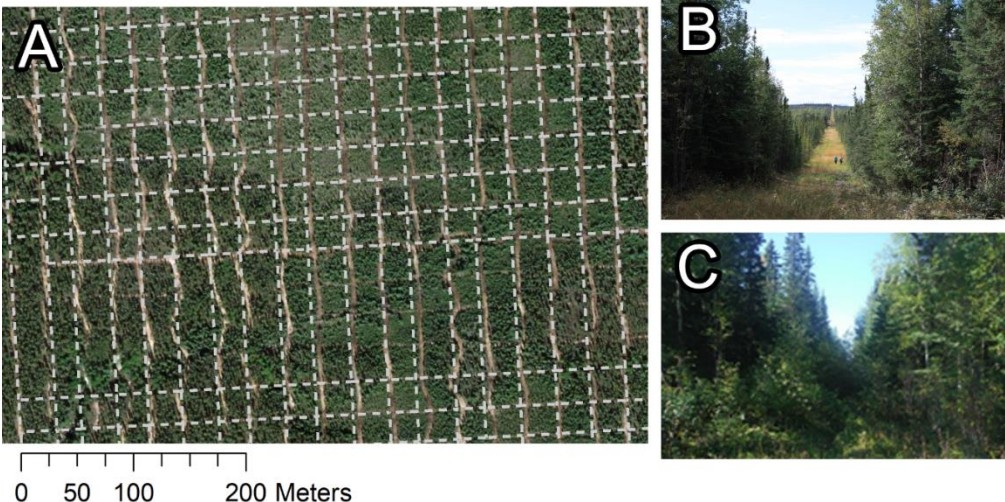

**Figure 1.** Seismic lines are linear-access trails cut through forests to allow the placement of geophones and other equipment associated with subsurface petroleum exploration. They occur widely in western Canada and are an example of linear disturbances that could benefit from refined mapping and attribution techniques. Existing public datasets created by manual digitization often contain spatial generalizations (dashed lines in (**A**)) and limited attributes. This makes it difficult to distinguish (for example) wide lines with little vegetation (**B**) from narrow lines with extensive regrowth (**C**). Image attribution: (**A**) Google Earth V7.3.3.7786 (5/22/2018); Alberta, Canada; 55°59′27.94″ N 111°10′36.68″ W; eye alt 1.65 km; Maxar Technologies 2020; accessed 3 November 2020; (**B**) Sarah Cole photograph; (**C**) Erin Bayne photograph.

Land managers and researchers require tools and workflows that permit the accurate delineation and attribution of linear disturbances in complex forest environments. Quackenbush [22] provided an overview of the basic approaches to automated linear feature extraction in her 2004 review. Classification/feature detection, particularly object-based approaches that employ spatial variables, may allow linear features to be mapped under some H-resolution conditions, where the pixel sizes are substantially smaller than the features of interest (e.g., [23–25]). However, the problem is notoriously complex. To address this challenge, many authors use mathematical-morphology operations and other transformations to enhance the appearance of linear features (e.g., [26–29]). Template matching uses similarity metrics to match image features to templates on the basis of spatial, spectral, or textural attributes [30–32].

Unfortunately, issues of occlusions and background noise in optical data regularly impede mathematical-morphology operations and template-matching techniques. The Hough transform converts features from image space to parameter space, wherein lines and other features can be represented as peaks or points in transformed space [33]. The Hough transform is more robust to small occlusions and noise than the original imagery and is widely used for linear-feature detection for mapping (e.g., [34–37]) and multi-image registration [38–40].

Despite the large volume of previous work, Hu et al. [41] noted that the linear-feature-detection problem is regularly ill-posed in complex scenes due to noisy or incomplete data. In forests, for example, shadows and occlusions from surrounding trees commonly hinder the spectral, spatial, and textural patterns that most optical techniques rely on. This is particularly true with narrow features such as trails and seismic lines. Complicating matters further, most linear-feature types in forests (trails, pipelines, seismic lines, transmission lines) have some sort of vegetative cover growing on the surface. In addition to exacerbating the occlusion issue, this further reduces the spectral contrast between the feature of interest and the surrounding matrix.

Recently, a number of studies have explored the use of LiDAR and photogrammetric data for linear-feature extraction in urban scenes (e.g., [42–44]). The height information contained in these three-dimensional datasets provides an additional source of information, and active sensors such as LiDAR are generally less susceptible to shadow, occlusion, and noise, which commonly hinder passive optical data [44]. Some authors have used intensity and height information from raw point clouds [45,46] or rasterized surface models [47] to extract linear features using simple classification techniques. For example, Soilán et al. [48] used heuristic rules and an unsupervised learning algorithm to extract roads in Dublin, Ireland with good accuracies.

Other authors have combined LiDAR and optical data to supplement the lack of multispectral information in most LiDAR datasets. For example, Liu et al. [49] fused multispectral data to LiDAR to map road networks in Bathurst, New South Wales with accuracies ranging from ~80–90%. Alternatively, LiDAR data can be rasterized and incorporated into pixel-based workflows. For example, Grote et al. [50] combined high-resolution multispectral data with height and intensity data from a LiDAR-derived digital surface model (DSM) to map road networks in suburban Grangemouth, Scotland and Vaihingen, Germany using image classification. Sameen and Pradhan [51] used a fuzzy ruleset in an object-based environment to perform urban road extraction with a similar DSM-multispectral dataset on the University Putra Malaysia campus in Selangor State, Malaysia.

To our knowledge, no previous studies have employed LiDAR data to extract linear disturbances from complex forest scenes. Linear-feature detection in forests provides a different challenge to road extraction in an urban setting. Forest scenes are extremely cluttered, and the linear disturbances are subtle and varied. However, the structural contrast between linear disturbances and surrounding vegetation provides patterns that could be exploited by workflows based on three-dimensional datasets.

Our approach to linear-feature delineation in forests uses a least-cost path (LCP) algorithm based on LiDAR-derived canopy height models (CHMs). Canopy height models represent the height of vegetation above local ground level and are derived by subtracting a digital terrain model (DTM) from a co-registered DSM [52]. Variations of the LCP strategy for tracing linear features have been widely used in medical imaging applications (e.g., [53–55]) and are similar to the shortest-path computations employed by Gruen and Li [30], Türetken et al. [56], and Wegner et al. [57] for urban/suburban road detection. The strategy is based on the notion that the most important attribute of a linear network is its connectedness [57], and that the LCP algorithm provides an effective means of tracing that connectedness between seed points located along the line. Since our implementation of the LCP workflow requires seed points provided by the operator, we describe our approach as semi-automated. However, seed points can also be derived from existing linear-feature databases, in which case the workflow can be fully automated.

The Forest Line Mapper (FLM) is an open-source Python software tool (v. 1.1; [58]), currently with ArcGIS dependencies, designed to map and attribute linear features in forests using an LCP approach

on CHMs. Our overarching goal was to evaluate the FLM toolset for mapping and attributing linear features in complex forest scenes with a variety of site types (upland/wetland) and stand characteristics. The specific objectives of this manuscript are:

1. To describe the algorithms adopted in each step of the FLM,
2. To report on an experiment comparing the accuracy of the FLM to a public dataset produced by manual digitization, and
3. To assess the potential of linear-feature metrics extracted by the FLM to characterize lines and ground conditions in a manner that might be useful for forestry and ecological applications.

To assist with our second and third objectives, we formulated a number of hypotheses based on expectations of how FLM outputs might compare to the best-available public dataset depicting linear features in our study area, plus a mechanistic understanding of how the resolution of input CHMs might influence FLM performance. Our hypotheses were: (i) that the semi-automated approach employed by the FLM would produce entities (centerlines and polygons) that are more accurate than publicly available datasets generated by manual photo-interpretation; (ii) that coarser-resolution input CHMs would reduce the accuracy of FLM outputs; and (iii) that high-resolution CHMs would provide enough information to attribute linear features with useful variables that could be used to characterize line types and ground conditions. We performed all of the tests on LiDAR-derived CHMs from boreal forest study areas located in the northeastern portion Alberta, Canada.

## 2. Materials and Methods

### 2.1. Overview and Study Structure

In this section we explain the line-mapping workflow we developed (Section 2.2), the application areas where we obtained reference field data (Section 2.3), and the statistical analyses we performed to assess accuracy and value of outputs (Section 2.4). The line-mapping tool we developed was named the Forest Line Mapper (FLM) and released as open-source code (v. 1.1; [58]) in Supplementary Material.

The workflow adopted in the FLM is based on the least-cost-path (LCP) algorithm, which calculates a weighted least-cost distance between two pixels in a raster image [59,60]. LCP is commonly used for habitat analysis [61,62] and urban planning [63,64]. However, it is also effective for delineating the path of linear features that are bound by distinctively higher or lower values in a raster image, such as drainage networks [65]. Hence, many linear forest disturbances can be mapped via associated canopy openings present in a CHM. In addition to the LCP, higher-cost paths can also be considered, up to a cost threshold, to form a least-cost-corridor (LCC) between two points. Our workflow incorporates the LCC to extract the areal footprint of linear features, in addition to their center line, in CHMs. Given a CHM, the center lines, and polygonal footprints of detected linear disturbances, the FLM then extracts size, shape, and height properties of the disturbances and incorporates them into the spatial database as attributes.

### 2.2. Linear-Disturbance Mapping Algorithm

The following subsections describe the algorithms adopted in the various tools available in the open-source toolset named the FLM. These tools were programmed in Python 2.7/3 (compatible with either) using Environmental Systems Research Institute (ESRI) ArcPy functions. A linear-disturbance mapping workflow chart is presented in Figure 2 and described fully in the following subsections.

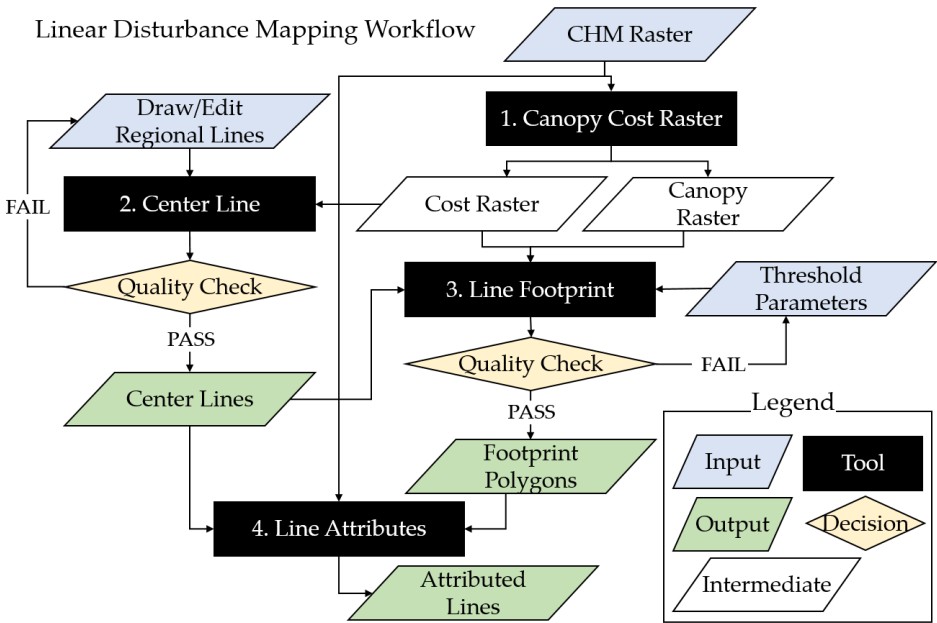

**Figure 2.** Flowchart of the semi-automated linear disturbance mapping workflow. The only inputs required from the user are a canopy height model (CHM) Raster, seed points, and threshold parameters. The latter two may require refinements in case outputs do not pass quality checks. Tools are fully automated python scripts listed in sequential order of which they are normally used. Tool numbers coincide with the 2.2 subsection numbers in which tool algorithms are explained. Optional parameters are available in each tool and not displayed in this graph.

FLM only requires two input datasets: (i) a CHM raster (Section 2.2.1); and (ii) regional seed points (Section 2.2.2) plus some information about line conditions in the form of threshold parameters (Section 2.2.3). All FLM tools are fully automated and allow for parameter tweaks, with default parameters being appropriate for most use cases. Users who aim for flawless outputs may require iterative trials and adjustments of parameter values and seed points, though users who are comfortable with minor errors may prefer to skip these intermediate quality checks.

FLM has three main outputs: (i) lines depicting the centerlines of linear disturbances; (ii) polygons depicting their extent; and (iii) line attributes. Some users may be interested in only obtaining FLM lines, which can act as refined versions of regional (i.e., lines digitized at ~1:20,000 scale) human-digitized linear feature databases. Other users may wish to obtain footprint polygons, which provide a detailed outline of footprint boundaries and can be useful in studying linear-feature conditions and micro-habitats. Finally, line attributes can provide managers and researchers with valuable information on disturbance conditions over large areas.

### 2.2.1. Inputs and Cost Rasters

LCP and LCC workflows require start and end points (i.e., seed points) for all linear features to be mapped, as well as a cost raster on which the cost-weighed distances are calculated. Seed points act as a rough guide of where individual features are located, and in the context of this study can be represented as the vertices of regional-scale linear-feature databases. Such databases are commonly found as publicly available linear-feature products. While these products may have limited accuracy at fine scales, their vertices are typically good enough to act as seed points for LCPs and LCCs. Raw CHMs can sometimes act as good cost rasters when mapping lines in dense forests. However, in the context of the boreal forest, sparsely vegetated terrain, especially on wetlands, causes LCPs derived from raw CHMs to cut corners between trees and LCCs to spread too far into the forests.

To create CHM-derived cost rasters appropriate for mapping linear disturbances in forests of varying levels of canopy density, we performed a series of treatments on raw CHMs. First, the CHM

was classified into "canopy" and "no canopy" classes by applying a height threshold to the CHM, arbitrarily set at 1 m. While this threshold worked well for our areas of interest (boreal forest of Canada), other thresholds may be required for other forest types. A reversed Euclidean distance raster (i.e., decreasing distance from source) was then calculated, using the canopy cells of the binary raster as source. This helps to guide the LCP to be positioned at the center of the linear disturbance feature. Additionally, we smoothed the binary canopy raster using a moving window, which calculates a normalized ratio between mean and standard deviation. The smoothed raster enhances the signal of canopy in sparsely vegetated forests, increasing the cost of gaps between trees, which helps to constrain LCCs to the boundaries of the disturbances. The original binary raster, its smoothed version, and the reversed Euclidean distance layer were then combined via user-defined weights. The FLM provides default weights based on our goals and experiences; users in other forest environments may need to adjust these values. Finally, the base exponential was calculated for each cell of the raster combination and raised to a user-defined power. This final step helps to increase the cost of vegetated terrain, and internal tests showed it increases consistency of outputs.

### 2.2.2. Center Line Mapping

Using the CHM-derived cost raster described above, polylines can be created at the center of linear disturbances by applying the LCP algorithm between two points, which we call "seed points". Seed points can be created by manually digitizing linear features as lines at moderate scales (~1:20,000) and exporting the vertices. Public GIS datasets that represent linear features as lines can also be used as a source of seed points. To reduce the memory usage and processing time, the cost raster was clipped around each line segment with a user-defined search radius before applying the LCP algorithm to segments. The cost-distance tool from ArcPy was used to calculate the least-cost distance between the starting point and all other pixels of the clipped cost raster. Finally, the LCP was found for each segment using the cost-path-as-polyline tool. To improve the performance on large datasets, which commonly have tens of thousands of line segments, a parallel-processing approach was taken on the steps mentioned above, such that multiple line segments could be processed simultaneously.

### 2.2.3. Linear Footprint Polygon Mapping

The software generates LCCs for each line segment in a similar fashion to the LCPs. First, the cost raster is clipped around the center lines with a user-defined maximum search radius. Then, two cost-distance rasters are generated for each segment: one using the start point as origin and another using the end point as origin. The two cost-distance rasters are summed, and then rescaled so that the minimum value (which relates to the cost of the LCP) becomes zero. This rescaled sum raster is referred to as the corridor raster.

Corridors can be mapped based on the corridor raster by applying a cost threshold to select disturbance cells. A least-cost corridor threshold (LCCT) can act as a global parameter, applied to all lines the same way. However, we found that linear-disturbance width and surrounding-forest density influence the appropriate LCCT value for each line. Therefore, these line conditions should be considered to define local LCCTs. We developed a tool to automate LCCT attribution based on canopy density surrounding lines, however it does not consider line width. For this reason, very wide lines (>10 m width) or very narrow lines (<3 m) should be manually set to global large and small LCCT values, respectively.

Once the LCC is mapped by selecting corridor raster cells with values smaller than the LCCT, cells classified as canopy in the binary canopy/no-canopy raster are removed from the footprint. Then, small artifacts and islands can be removed via optional cell-erosion procedures. Finally, footprint cells are converted to polygons and merged into the final product. All procedures mentioned for linear-footprint mapping are performed in parallel to maximize performance.

### 2.2.4. Line Attribution

Given the input CHM, output center lines, and footprint polygons, it is possible to extract a series of attributes that may be useful in classifying and assessing the conditions of linear disturbances. In the context of boreal restoration, line width, orientation, and microtopography can be important factors on the microclimate [13] and growth success of newly planted seedlings [10,66]. Therefore, the FLM attributes center lines with spatial information about their length, bearing, direction, and sinuosity. Then, based on their respective polygon footprints, lines receive the following additional attributes: total footprint area, perimeter, average width, and perimeter/area ratio (PAR). Finally, based on the CHM cells within their footprint, the lines receive the following height attributes: average height, total CHM volume, and root-mean-square height (RMSH). Table 1 describes each of the FLM attributes and their units of measure.

**Table 1.** Forest Line Mapper (FLM) attributes described with units of measurement. Each forest line mapped with FLM can receive these attributes. Some attributes are derived from polylines of the forest lines, some from footprint polygons surrounding the forest lines and some attributes are derived from canopy CHM cell values within those surrounding footprint polygons.

| Line Attribute | Description | Units |
| --- | --- | --- |
| Length | Line length from start to end of segment | meters |
| Bearing | Clockwise angle between north and line joining first and last segment vertices | degrees |
| Direction | Quadrant direction closest to line bearing | text label |
| Sinuosity | Line length divided by Euclidean distance from start to end of segment | unitless |
| Area | Area of footprint polygon surrounding line | meters squared |
| Perimeter | Perimeter of footprint polygon surrounding line | meters |
| Average Width | Average width of footprint polygon surrounding line | meters |
| Perimeter/Area | Perimeter-area ratio of footprint polygon surrounding line | $meters^{-1}$ |
| Average Height | Average height of CHM cells within surrounding footprint polygon | meters |
| Volume | Area of CHM cell multiplied by sum of CHM values within surrounding footprint polygon | cubic meters |
| RMSH | Root-mean-squared height (RMSH) of CHM cell values within surrounding footprint polygon | meters |

Line attribution is performed via an automated tool that first segments the lines based on user preference (whole lines, line intersections, arbitrary length) then attributes the segments (Figure 3) via parallel processing. Additionally, the FLM is capable of extracting and summarizing information from external raster images as attributes on forest-line segments. For example, the attributes in Figure 3 were summarized from a user-supplied external raster showing the spatial distribution of coarse-woody debris (CWD) across one of our areas of interest (see Queiroz et al. [67] for details). CWD is an important element of linear-feature restoration in the boreal forest, providing microsites for regenerating seedlings and restricting the movement of humans and predators along lines. The FLM will summarize any user-supplied external raster in a similar manner.

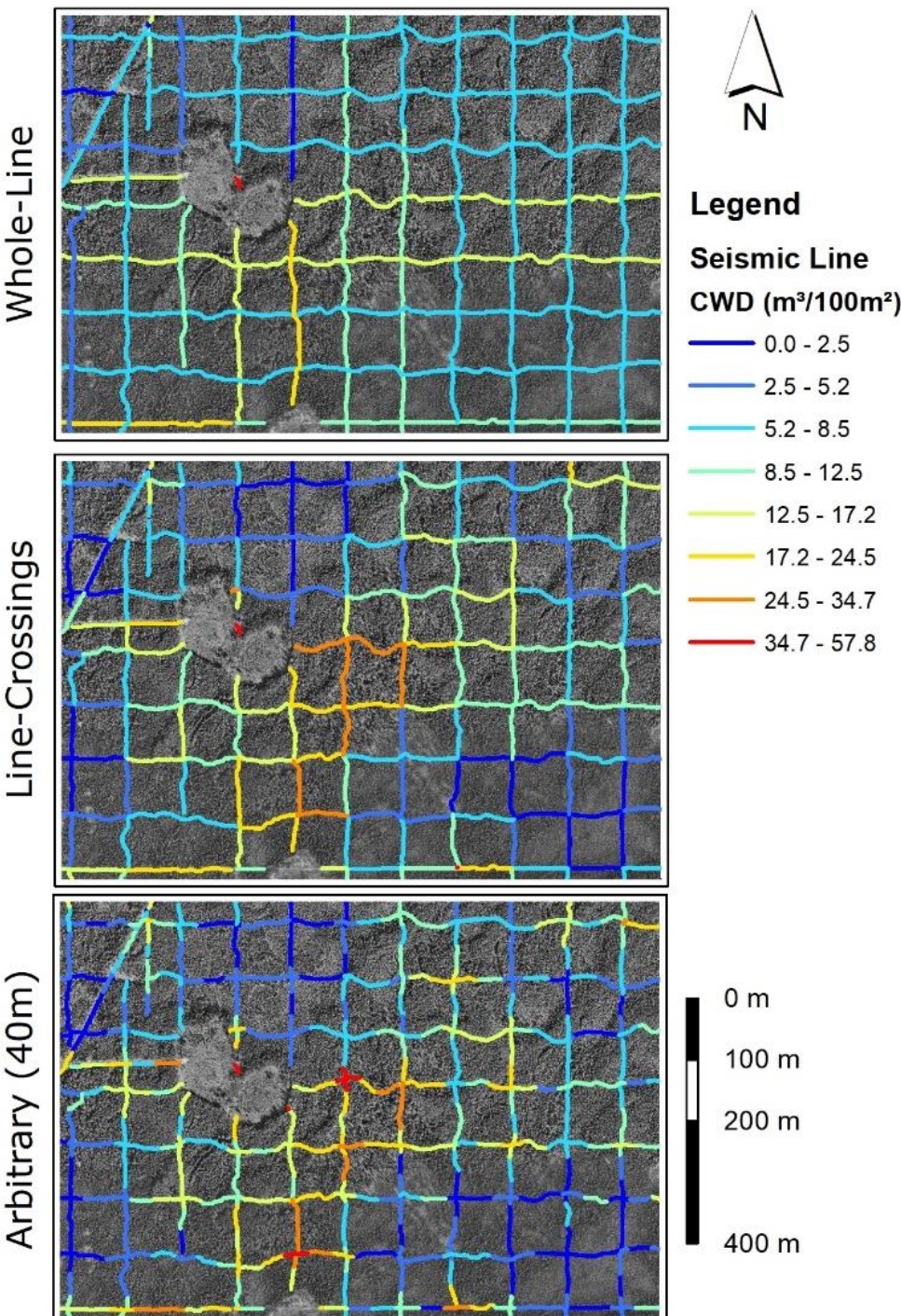

**Figure 3.** The Forest Line Mapper's three segmentation options for defining the entities used for summarizing attributes: whole-lines, line-crossings (line intersections), and arbitrary. The whole-line option summarizes attributes across the entire line. The line-crossing option summarizes attributes between intersections. The arbitrary option summarizes attributes along arbitrary (in this case, 40 m) segments. These three example maps show average coarse woody debris (CWD; fallen dead trees) volume on seismic lines and were derived from a CWD raster layer from a separate study (Queiroz et al., 2020).

## 2.3. Study Area

To test the accuracy and effectiveness of the FLM, we applied it to three study sites in northeastern Alberta's boreal forest (Figure 4). These areas are referred to as Kirby (416 ha in size), LiDEA I (828 ha), and LiDEA II (9972 ha; n.b. LiDEA is an acronym for Cenovus Energy's Linear Deactivation project). The study sites are located in the central-mixedwood subregion of the boreal forest. The central mixedwood is characterized by gently rolling terrain partitioned into forested uplands, composed mostly of jack pine (*Pinus banksiana* Lamb.), black spruce (*Picea mariana* (Mill.) B.S.P.), and trembling aspen (*Populus tremuloides* Michx.) trees, and wetlands composed of black spruce and tamarack (*Larix laricina* (Du Roi) K. Koch) forests with shrubs, sedges, and moss understory [68].

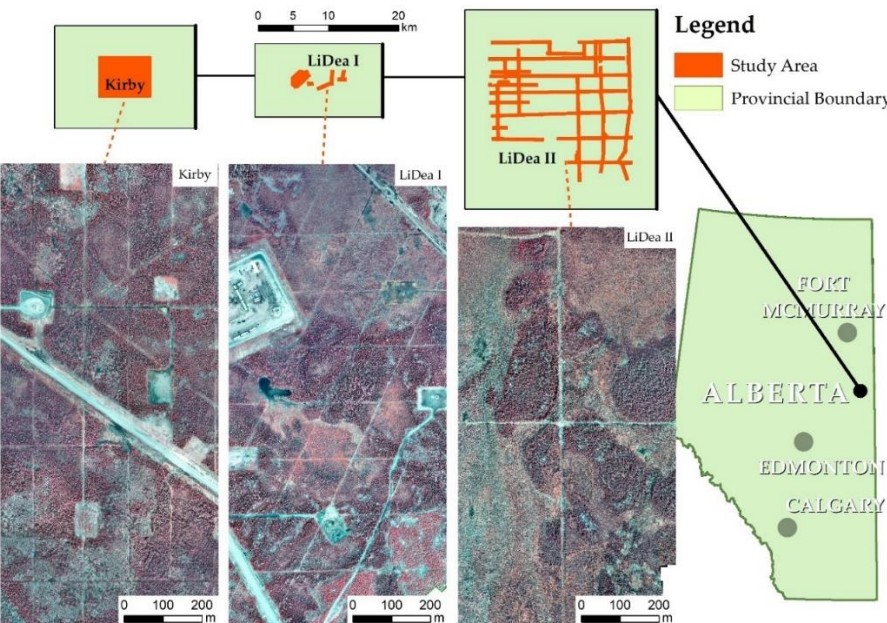

**Figure 4.** Display of the study sites Kirby, LiDEA I and LiDEA II in northeastern Alberta. Each area is presented with an inset at a larger scale showing a false color aerial photo of example linear disturbances. Airborne imagery is the property of the Boreal Ecosystem Recovery and Assessment (BERA) project.

The study sites are covered with different densities and types of linear disturbances, including from narrowest to widest: low-impact and legacy seismic lines, roads, pipelines, and power lines. The terms "legacy" and "low-impact" seismic lines indicate: (i) the era in which the lines were constructed; and (ii) the resulting physical characteristics of the disturbance. Early legacy surveys involved clearing vegetation along 5- to 10-m wide corridors using bulldozers. Low-impact surveys, which began around 2000 in Alberta, involved narrower corridors (~2- to 4-m wide) cleared using mulchers and other light equipment. Readers interested in further information on seismic lines are referred to Dabros et al. [4].

LiDEA II has the lowest density of disturbances, being a sparse grid of legacy lines. LiDEA I has the densest low-impact grid, road networks, and pipelines. Kirby has legacy and low-impact lines, roads, and a very wide powerline disturbance. Both Kirby and LiDEA I have wellpads, cutblocks, constructions, and other non-linear clearings that were not addressed in this study. The study areas provide a variety of western-boreal forest stands and wetlands affected by linear features of different widths and in different densities.

### 2.3.1. Remote Sensing Data

Dense airborne LiDAR data (~40 pts/m$^2$) was obtained over the three sites by OGL Engineering of Calgary, AB, Canada during August of 2017. The platform used to obtain this data was a Cessna

210T aircraft, which flew at 67 m/s ground speed and at 850 m above ground carrying a LiDAR sensor (Leica ALS70, Leica Geosystems AG, Heerbrugg, Switzerland), an optical sensor (Leica RCD30, Leica Geosystems AG, Heerbrugg, Switzerland), a global navigation satellite system (GNSS, Rohde & Schwarz, Munich, Germany) unit, and an inertial measurement unit. OGL Engineering performed noise removal, georeferencing, and ground classification of the LiDAR points using Bentley Microstation with Terrasolid software. We then rasterized the ground points into a digital terrain model (DTM) and the first-return points into a digital surface model (DSM) in ESRI ArcMAP (Version 10.7.1) using 25 cm grid cells. Finally, a CHM was obtained by subtracting the DTM from the DSM. We resampled the 25 cm ground-sampling-distance (GSD) CHM into five additional CHMs of 50 cm, 1 m, 2 m, 4 m, and 8 m GSD to simulate coarser products that could be obtained from sparse LiDAR (Section 2.4). These CHMs were used as FLM inputs to map linear features in the study areas and test the effect of spatial resolution on output quality (our second hypothesis).

### 2.3.2. Reference Data

As reference data for our accuracy assessments associated with our first two hypotheses, we used field-measured seismic line locations obtained during a stocking survey performed as part of the Boreal Ecosystem Recovery Assessment (BERA; www.bera-project.org) project. A total of 59 sites were selected from the three study areas across gradients of line width (legacy and low-impact lines) and orientation (north-south, east-west, and diagonal). Each site was comprised of a 150-m stretch of seismic line, within which detailed field surveys measuring seismic-line centers and widths were made in the summer of 2017.

Centerline Location Reference Samples

We randomly selected 245 points from the field survey (at least 10 m apart from one another) on the seismic lines across our study sites to measure the location of the centerline. To do this, a surveyor visited each point and laid out a rope perpendicular to the line connecting two edges. Then, they located the center point of the rope using a tape and measured the X, Y, and Z coordinates of that center point using a Real Time Kinematic (RTK) GNSS system (Trimble R8—8 mm horizontal and 15 mm vertical precision-Trimble, Sunnyvale, USA). The surveyor also recorded the general width of the line by averaging a few (3–5) measurements within 10 m of the points. 100 of these points were located on legacy lines and the rest (145) were located on low-impact lines. Kirby, LiDEA I, and LiDEA II had 93, 81, and 71 sample points, respectively.

Line Width Reference Samples

We randomly selected 103 points at least 10 m apart from one another on the seismic lines across our study sites to measure the width of the disturbance. To do this, a surveyor visited each point and laid out a rope perpendicular to the line connecting two edges. They then used a tape to measure the width of the line and used a RTK GNSS system (Trimble R8) to measure the X, Y, and Z coordinates of the edge points. 48 of these points were located on legacy lines (mean width 7.15 m with a range of 4.72–10.45 m and standard deviation of 1.24 m) and the rest (55) were located on low-impact lines (mean width 4.50 m with a range of 3.249–6.70 m and standard deviation of 0.65 m). LiDEA I and LiDEA II had 55 and 48 sample points, respectively.

### 2.4. Accuracy and Statistical Analyses

To assess the accuracy of the proposed workflow relative to the resolution of the input CHM raster (our second hypothesis), we performed a series of tests. To create base layers for these tests, we resampled a high-resolution (25 cm pixel-size) CHM raster using bilinear sampling and iteratively doubling the cell size to obtain coarser rasters (Figure 5). We assume that, for the purposes of these tests, the resampled rasters are suitable representations of CHMs generated from LiDAR point clouds of different point densities. We applied the FLM workflow to each resampled raster to obtain centerlines

and footprint polygons (Figure 5) attainable at different CHM resolutions. Then, we performed spatial accuracy tests of the centerline locations and footprint polygon widths using field data of Kirby, LiDEA I, and LiDEA II for reference, which are explained in the following subsections.

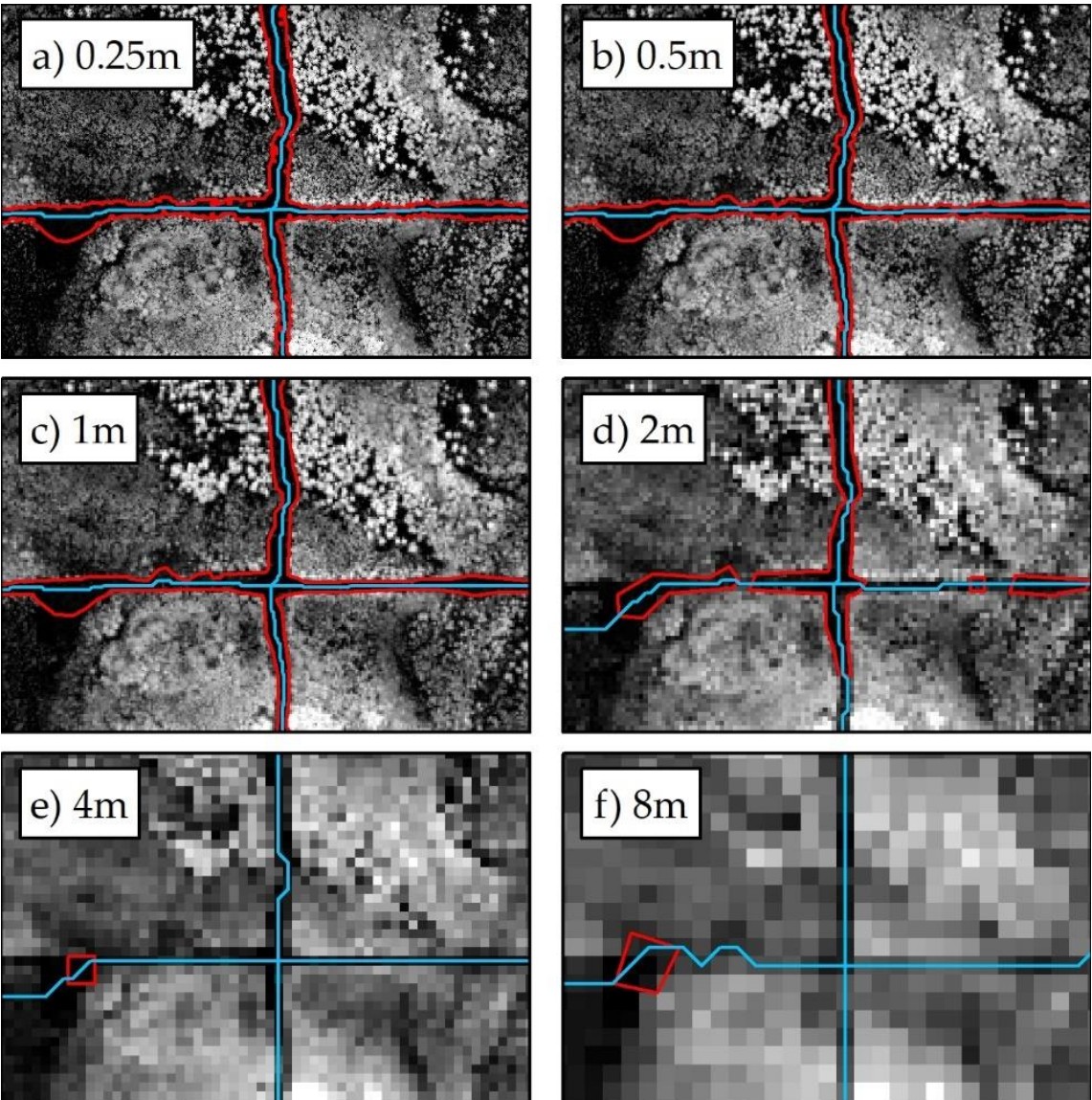

**Figure 5.** Output centerlines (blue) and footprint polygons (red) of legacy seismic lines mapped with the proposed software tool overlaid on CHM raster resampled from 25 cm (**a**) resolution to 50 cm (**b**), 1 m (**c**), 2 m (**d**), 4 m (**e**), and 8 m (**f**).

Even though the FLM mapped all linear disturbances across the three study areas, accuracy tests were performed only on seismic lines (legacy and low-impact), which were the focus of our field efforts. Seismic lines are the narrowest linear disturbances on the landscape, and we therefore assume the most difficult to map. In addition to absolute errors, we also report our accuracy results relative to line width, and therefore assume that they represent worst-case scenarios of what might be expected on wider, more-prominent features such as roads, pipelines, and powerlines.

Using the highest-resolution CHM (25-cm pixel size) and its respective output layers, we attributed the seismic lines of Kirby (2.2.4) and performed multivariate analysis of variance (MANOVA) of line groups to test our third hypothesis, as explained in 2.4.2. Centerline and footprint accuracy assessments

were performed in ArcGIS (Version 10.7.1), and line attribute MANOVA was performed in R (version 3.5.1; R Core Team, 2018).

### 2.4.1. Centerline Spatial Error Assessment

Centerline location samples (Section 2.3.2) were used as reference points to assess the accuracy of centerlines generated by the FLM. We measured the shortest distance from a reference point to its corresponding line from the FLM. The absolute difference between field-measured and FLM-predicted centerline locations was calculated using Equation (1):

$$MD = \frac{\sum_{i=1}^{i=n} Dir}{n} \tag{1}$$

where *MD* is the absolute mean deviation of centerline measured in m, *Dir* is the vertical distance from reference center point *r* to FLM-generated centerline *i*, and *n* is the total number of samples. A second relative measure of mean deviation, normalized by line width, was calculated using Equation (2):

$$MD\ (\%) = \frac{\sum_{i=1}^{i=n} \frac{Dir}{Wi} * 100}{n} \tag{2}$$

where *MD* (%) is the percent mean deviation of centerline and *Wi* is the general width of line *i* as measured in the field.

We repeated these calculations for centerline products generated using CHM inputs of different resolutions (25, 50, 100, 200, 400, and 800 cm). Finally, we reported overall deviation (absolute and percent) and deviations for legacy and low-impact lines separately. In order to test the first element of our first hypothesis, we compared the accuracy metrics of the FLM products with those from the best publicly available datasets [69], generated through manual photo-interpretation using z tests.

### 2.4.2. Line Footprint Width Assessment

Line-width samples (Section 2.3.2) were used as reference data to assess the accuracy of polygons delineated by the FLM, which we equate to estimates of footprint width. We first assessed if the footprint of a field-measured line was detected at all by the FLM. This was done by visually inspecting each reference (line) sample overlaid on the corresponding footprint polygon layer in ArcMap (Version 10.7.1), then calculating the detection rate using Equation (3):

$$DR = \frac{Nd}{n} * 100 \tag{3}$$

where *DR* is the detection rate; *Nd* is the number of lines detected by the proposed workflow; and *n* is the total number of sample lines. If a line was not detected, we assigned zero (0) to the width value of that line. If a line was detected, we calculated its predicted width from the output footprint layer. To do so, we calculated the mean footprint polygon width in a 12.5-cm buffer around the field measurement. Then, the absolute difference between field-measured and FLM-predicted line widths was calculated using Equation (4):

$$MAE = \frac{\sum_{i=1}^{i=n} |(Wir - Wio)|}{n} \tag{4}$$

where *MAE* is the mean absolute error measured in m, *Wir* is the width of a line *i* as measured in the field (reference width), *Wio* is the width of line *i* from the FLM output footprint layer o; and *n* is the

number of samples. A second relative measure of mean absolute error, expressed as a percentage of line width, was calculated using Equation (5):

$$MAE\ (\%) = \frac{\sum_{i=1}^{i=n} |\frac{(Wir - Wi)}{Wir}| * 100}{n} \tag{5}$$

where *MAE* (%) is mean absolute error expressed as a percentage of line width.

We repeated these calculations for polygon products generated using CHM inputs of different resolutions (25, 50, 100, 200, 400, and 800 cm resolution). In order to test the second element of our first hypothesis, we compared the accuracy metrics of the FLM products with those from the best publicly available datasets [69], generated through manual photo-interpretation using z tests.

### 2.4.3. Line Treatment Types MANOVA

A portion of the seismic lines in the Kirby study area received restoration treatment in 2015, including silvicultural treatments and planting of native seedlings. On wetland areas, where the water table is close to the surface, mounding is a common treatment that enhances the survival rate for newly planted seedlings by creating microsites that are relatively drier than their surroundings [10,70]. On upland areas, CWD are commonly added either in piles or across the entire extent of the seismic lines [71]. Mounding and CWD are applied with the dual purposes of enhancing the survivability of seedlings as well as reducing line use by wildlife and humans [71]. Lines with mounding and/or CWD treatment were visually identified and labeled on aerial images of the study area, with the aid of a 2015 line-treatment proposal layer. The line-treatment proposal layer was helpful as a general guide to identify treatments, but was not always reliable given field conditions (wetland/upland) encountered by operators also dictated the final treatment type. Consistent site-wide treatment information could not be obtained in LiDEA I or LiDEA II.

In order to test our third hypothesis, that high-resolution CHMs would provide enough information to characterize line types and ground conditions, we labeled lines based on size and treatment: legacy and low-impact seismic lines (legacy lines wider than 5 m and low-impact lines narrower than 5 m, respectively). Each line was then subdivided into treatment subgroups: untreated, mounded, and CWD. All lines were divided into 100 m segments for the purpose of this classification. Extremity segments shorter than two standard deviations from the mean (in meters) were removed from the analysis. These segments were attributed using FLM (Section 2.2.4), producing attributes of size, shape, and CHM variables. Line-segment attribute values were incorporated as continuous dependent variables grouped by the line labels (independent variable) in a multivariate analysis of variance (MANOVA). Our alternate hypothesis was that the variance between groups would be greater than the variance within groups ($\alpha = 0.01$). In other words, different line categories and treatments would produce different attribute signatures when studied using the proposed software tool.

## 3. Results

### 3.1. Linear-feature Mapping Performance

Figure 6 shows all linear disturbances mapped across the three study areas: Kirby, LiDEA I, and LiDEA II. Footprint polygons were generated for both seismic lines as well as wider linear disturbances. Center lines were generated only for seismic lines. The total length of seismic lines and footprint area, as well as the processing times for each project area, are summarized in Table 2. Altogether, the FLM mapped and attributed more than 675 km of linear features with a processing time of just over 47 min.

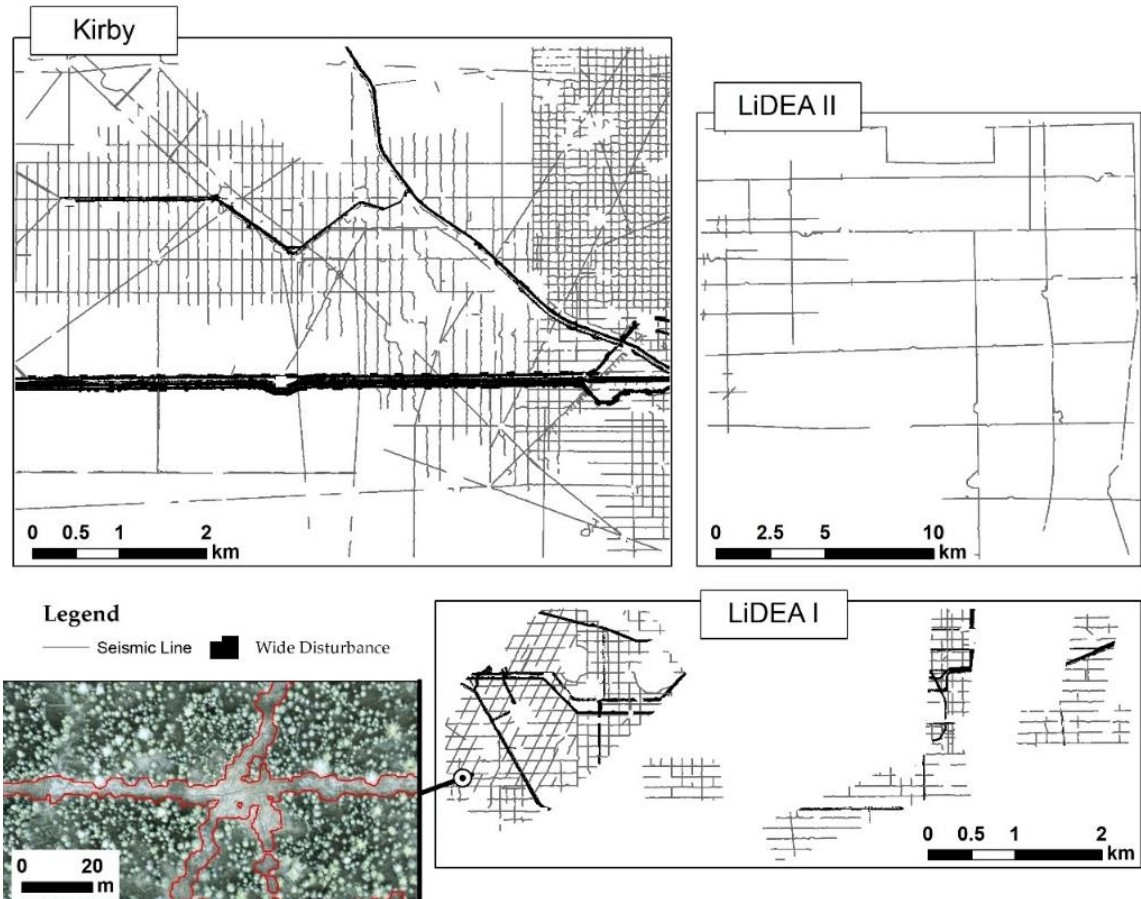

**Figure 6.** Mapped linear features (seismic lines and wide disturbances) using 25 cm pixel-size CHM inputs in the study areas Kirby, LiDEA II, and LiDEA I. The latter includes an inset map shown at larger scale with footprint polygons outlined in red overlaying an orthophoto background. Wide disturbances include roads, pipelines, and power lines.

**Table 2.** Numeric results of mapping seismic lines (SL) and wide linear disturbances (WLD, i.e., roads, pipelines, power lines) using 25 cm pixel-size inputs. Processing time is given for the center-line step on seismic lines, which is the same order of magnitude as for the footprint step, obtained using Python 2.7.16 (34-bit) with 8 processing cores on an Intel Core i7-2600 CPU at 3.40 GHz.

|  | SL Length (km) | SL Footprint (ha) | WLD Footprint (ha) | Surveyed Area (ha) | Disturbance Area (%) | Processing Time (minutes) |
|---|---|---|---|---|---|---|
| Kirby | 344.66 | 130.69 | 176.05 | 4,416.30 | 7 | 23.2 |
| LiDEA I | 86.25 | 37.44 | 54.77 | 827.88 | 11 | 8.7 |
| LiDEA II | 244.72 | 142.47 | 0.00 | 9,972.06 | 1 | 15.5 |

LiDEA I was the densest area in terms of linear disturbances, followed by Kirby and LiDEA II. The latter did not have any wide linear disturbances except seismic lines. The total footprint area of seismic lines in LiDEA II was larger than that of Kirby, even though the latter had larger total seismic line length, given LiDEA II had mostly legacy lines and Kirby had a mix of low-impact and legacy lines.

Seismic lines in Kirby were attributed with length, bearing, orientation, sinuosity, area, average width, perimeter, perimeter-area ratio (PAR), average vegetation height, vegetation volume and vegetation roughness. PAR showed interesting spatial patterns and is presented in Figure 7. Wide legacy seismic lines, which are mostly positioned NE-SW or NW-SE, displayed low PAR except where there was CWD treatment. Low-impact lines, which are mostly positioned N-S or E-W, displayed low PAR on wetlands and high PAR on upland forest stands.

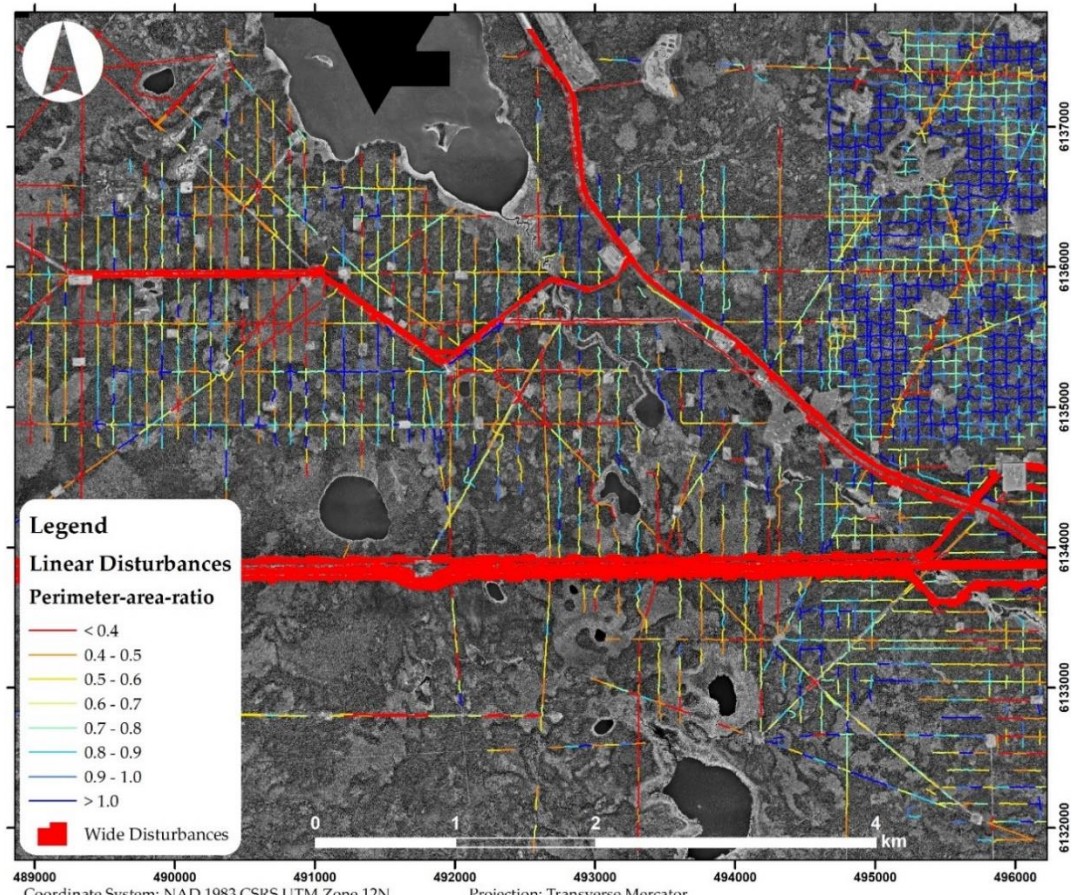

**Figure 7.** Mapped linear features in Kirby using 25cm pixel-size inputs. Seismic lines are classified by perimeter-area-ratio (PAR). Wide disturbances include roads, pipelines and power lines.

## 3.2. Accuracy Assessments

### 3.2.1. Centerline Spatial Error

Figure 8 presents the percent mean deviation of predicted line center from field measured center as absolute unit (A), and relative to the seismic line width (B), for different input CHM resolutions. We focus our observations on the relative measures of error since they account for the size of the physical disturbance so are more insightful.

As expected, mean deviation increased as the spatial resolution of the input CHM resolution decreased, supporting our second hypothesis. Also, low-impact lines exhibited higher mean deviation and legacy lines exhibited lower mean deviation consistently across resolutions. This might be because low-impact lines are narrow, complex in shape and difficult to map, whereas legacy lines are generally wider, straight, and therefore easier to delineate. Accuracy did not vary remarkably up to 200 cm input resolution, and mean deviation remained well below 20% of line width. The best-case scenario for legacy and low-impact seismic lines was 6.44% (0.46 m) and 11.02% (0.44 m) deviation, respectively, with 25 cm resolution inputs. In contrast, publicly available line datasets had average deviation over 60% for our study sites. Statistically, FLM centerlines were significantly more accurate ($p < 0.01$) than publicly available lines when the input CHM resolution was of 4 m or less. These results support our first hypothesis–that the FLM is more accurate than publicly available datasets produced through manual digitizing.

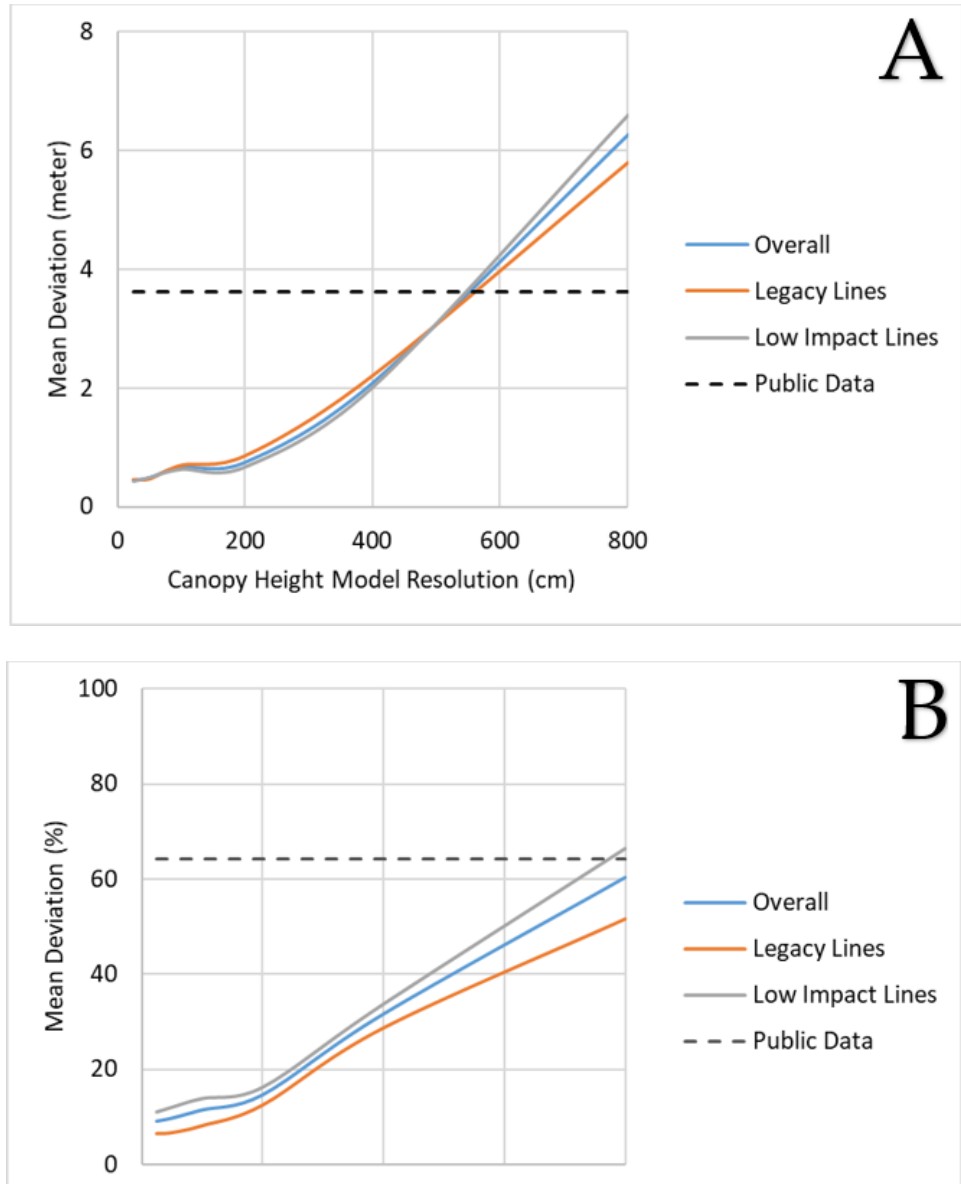

**Figure 8.** Mean deviation of mapped centerlines from field-measured center points expressed in absolute units (**A**) and percentages relative to line width (**B**) for different input CHM resolutions. Dashed lines show the accuracy of the best publicly available linear-feature datasets in our study sites.

3.2.2. Line Footprint Width

Figure 9 presents the MAE of predicted line in absolute unit (A) and relative to field measured line width in percent (B) using different input CHM resolutions. Once again, we focus our observations on the relative metrics.

In accordance with our second hypothesis, lower MAE was observed at higher CHM resolution (lower pixel size) in most cases. The best-case scenario for legacy and low-impact seismic lines was 17.27% (1.19 m) and 27.41% (1.21 m) MAE respectively with 25 cm resolution inputs. Overall, MAE (Figure 9-blue trendline) and low-impact line MAE (Figure 9-grey trendline) exhibited a sharp increase after 100 cm CHM resolution, whereas legacy lines maintained low MAE up to 200 cm resolution and then exhibited a sharp rise. This is due to the fact that low-impact lines are generally narrow and

thus a CHM that is coarser than 1 m resolution cannot be used to accurately map the linear footprints, whereas legacy lines are generally wide and thus even a 2 m CHM can be used to map their footprint. Therefore, in the case of line widths, these results only support our first research hypothesis when using high-resolution (<1-m pixel size) inputs.

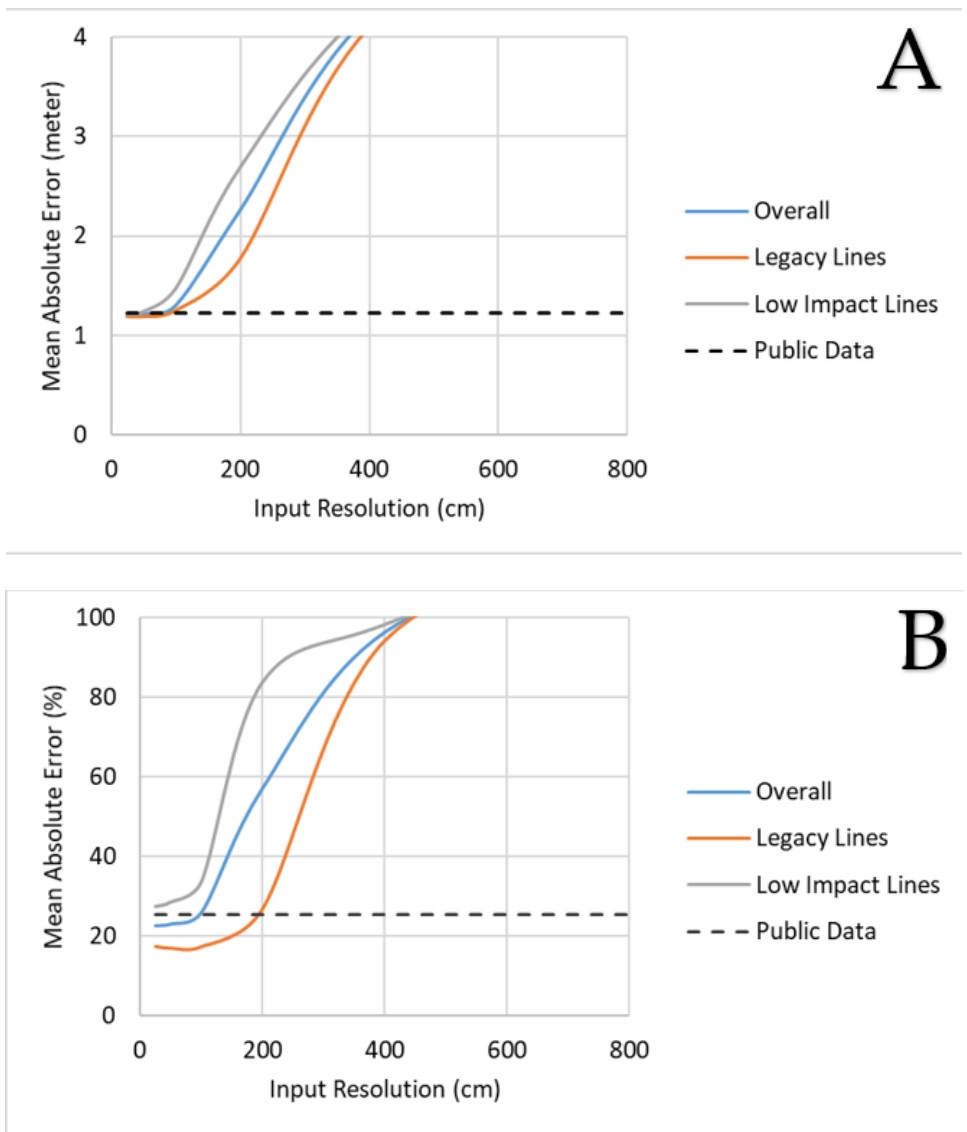

**Figure 9.** Mean absolute error (MAE) of predicted line width with reference to field-measured line width for different input CHM resolutions. Charts express errors in absolute units (**A**) and percentages relative to line width (**B**). Dashed lines show the accuracy of the best publicly available linear-feature datasets in our study sites. Note that while width-error levels could exceed the values presented here, measurements were confounded by situations where line spacing began to approach 2x the resolution of the CHMs being tested. As a result, we limit our recording of errors here at 100% MAE (4 m).

### 3.3. Line Treatment Types ANOVA

Lines were grouped based on size classes (legacy lines and low-impact lines) and treatments (untreated, CWD, mounding), presented in Table 3. The MANOVA results suggest that the variance between groups was significantly greater than within groups with a $p$ value smaller than 0.001. Similarly, the ANOVA of each individual factor (sinuosity, PAR, average width, average height, roughness) also rejected $H_0$ with $p$ values smaller than 0.001. To visualize the variance of the dependent variables between groups we created boxplots of each MANOVA factor (Figure 10).

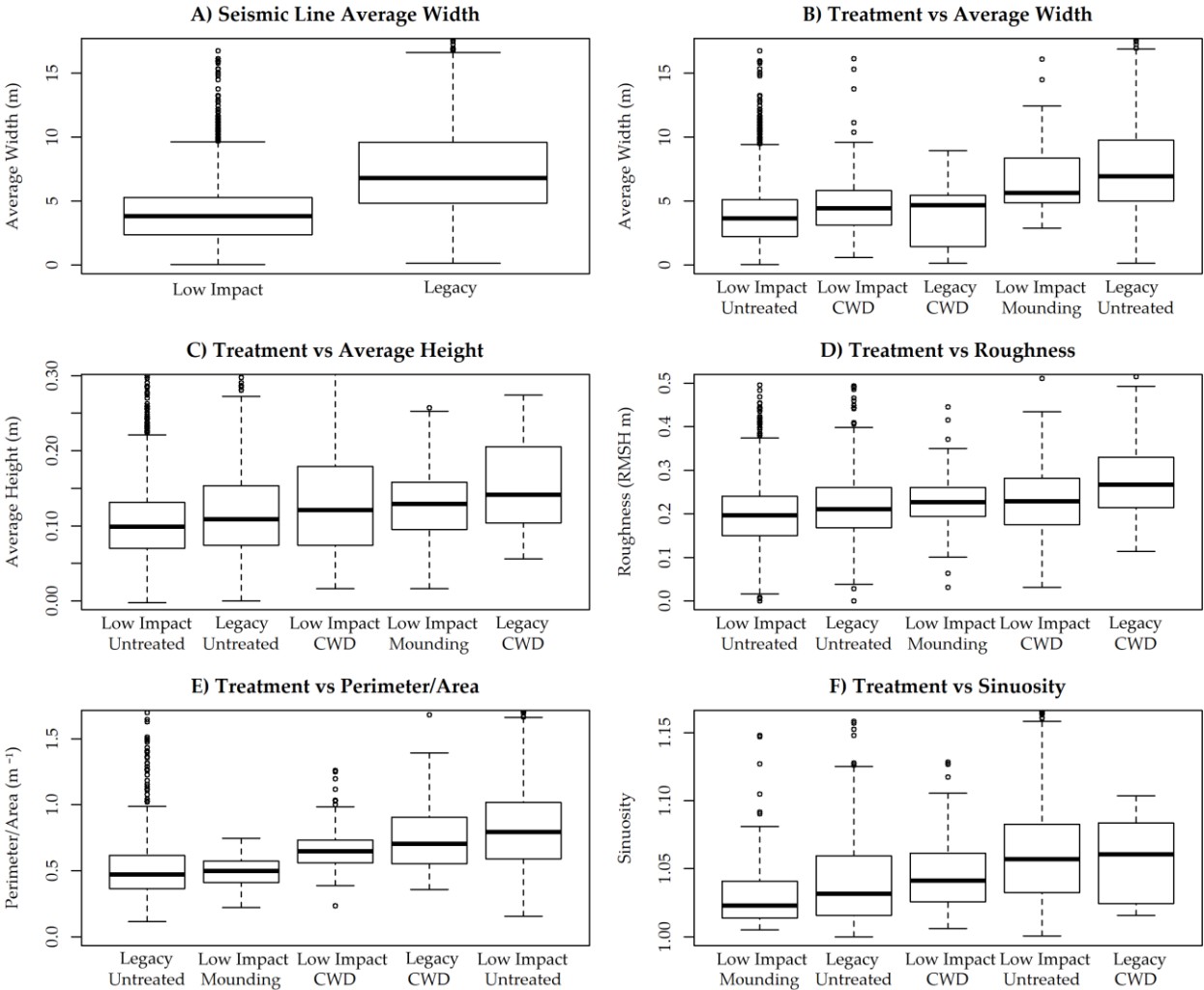

**Figure 10.** Box plots of dependent variables per line classes in the Kirby study area. Dependent variables include: average width (**A**,**B**), average height (**C**), roughness ((**D**), root mean square height, RMSH), perimeter/area (**E**), and sinuosity ((**F**), unitless).

**Table 3.** Seismic lines were grouped based on size and treatment classes. Size classes include legacy and low-impact lines. Treatment classes include untreated, CWD, and mounding. * Legacy Mounding lines were discarded due to their small sample size.

| ID | Group | Sample Size |
| --- | --- | --- |
| 1 | Legacy CWD | 38 |
| 2 | Legacy Mounding * | 14 |
| 3 | Legacy Untreated | 734 |
| 4 | Low-impact CWD | 165 |
| 5 | Low-impact Mounding | 95 |
| 6 | Low-impact Untreated | 2024 |

Line width was, as expected, much greater on legacy lines than low-impact lines; it was mostly above 5 m for the former and below 5 m for the latter. Low-impact Mounded lines presented large width and average height, as well as lower PAR and sinuosity when compared to untreated low-impact lines (162%, 124%, 61% and 54% on average, respectively). Lines with CWD presented large average height and roughness, as well as small width and PAR when compared to untreated lines (123%, 118%, 97% and 91% on average, respectively). Lines with CWD were distinct from lines with mounding relative to sinuosity, perimeter/area and average width (166%, 140% and 73% on average, respectively). These results not only show statistical significance but also conform with expectations for different types of forest lines and ground conditions, therefore supporting our third hypothesis–that CHM-derived line attributes can provide valuable information about forest line characteristics.

A few tests were made comparing predicted average line width between groups of lines separated by treatment, size and terrain type (upland vs. lowland). Upland lines were narrower than lowland lines (average 4.47 and 7.14 m, respectively), regardless of treatment, both for legacy and low-impact groups. Low-impact upland lines were the narrowest type even without treatment (avg. 3.55 m). Low-impact mounded lines (avg. 6.61 m) were narrow when compared to legacy untreated lowland lines (60% average width) but wide when compared to low-impact untreated lowland lines (128% average width). CWD lines (avg. 4.61 m) showed a similar pattern when compared to their legacy and low-impact untreated upland counterparts (67% and 144%, respectively).

## 4. Discussion

We tested three research hypotheses in this study, each of which were corroborated by our results. The first was that FLM outputs would be more accurate than the best publicly available linear-features dataset in the region [69] in the task of delineating linear-feature center lines (polylines) and footprints (extent polygons). We observed significant differences ($p < 0.01$ for z tests) in centerline accuracy for all but the coarsest-resolution (800-cm pixel size) CHM we tested, with the FLM performing better than the public dataset produced by human photo-interpreters. The difference was less pronounced when measuring seismic line width, but still present in high-spatial-resolution (<1m pixel size) CHM data.

Our second hypothesis was that coarser-resolution input CHMs would reduce the accuracy of FLM output lines and polygons. We observed strong direct relationships between error levels and CHM resolution, both for center-line location and footprint widths.

Finally, we hypothesized that high-resolution CHMs would provide enough information to attribute lines with variables that would be useful for characterizing line types and ground conditions. We observed statistically significant ($p < 0.001$) differences in FLM attributes between lines grouped by both size class (legacy and low-impact) and treatment status (untreated, CWD, and mounding), in addition to statistically significant ($p < 0.001$) differences between each individual FLM attribute (sinuosity, PAR, average width, average height, and roughness).

### 4.1. Contributions to Linear Disturbance Mapping Literature

Our work contributes substantially to the linear-disturbance mapping literature and is the first study, that we are aware of, to employ LiDAR CHMs to extract linear disturbances in complex forest scenes. As Wegner et al. [57] stated, the most important attribute of a linear network is its connectedness, and the LCP approach used by the FLM has proven to be an effective strategy for delineating and attributing even subtle linear disturbances like seismic lines. LiDAR-derived CHMs can effectively capture the structural contrast between linear disturbances and surrounding vegetation and present a good foundation for mapping linear features.

Linear disturbance mapping with the FLM software tool takes some practice and effort but can produce layers of excellent quality and detail across extensive application areas. Given that 1-m resolution LiDAR-derived CHMs are now becoming common in many jurisdictions, we expect that the proposed workflow will be useful in future regional assessments of linear disturbances in many forest types.

The capacity of the FLM to generate valuable information products over large areas is expected to provide significant aid to researchers and resource managers working in forested environments. For example, the woodland caribou (*Rangifer tarandus caribou*) is a threatened species in Alberta whose decline has been linked to habitat fragmentation [71]. In our study areas, the FLM was able to identify linear features with little or no vegetation or CWD, which serve as barriers to caribou predators and humans [72] and might therefore benefit from targeted restoration efforts.

### 4.2. Advantages of Semi-Automated Approach

The FLM yields accurate, consistent, and extensive outputs that are either not feasible or unattainable via manual delineation or fully automatic detection. Manual delineation can be either accurate or extensive, but not both without contributing exceptional resources to the task. Fully automated methods available in the literature commonly aim to be extensive while suffering from large commission and omission errors, especially on narrow features such as low-impact seismic lines [73]. A semi-automated approach used by the FLM capitalizes on the analytical power of the human brain in conjunction with the raw processing power of modern computers. For any given application area, we expect the FLM approach to require longer setup time but produce substantially more accurate and insightful outputs than either manual or fully automated approaches.

We are not aware of other methods in the literature that enable extensive delineation and attribution of linear features. We expect line attribution to be useful in studying vegetation growth trajectories, human and wildlife line use, and managing line restoration projects.

Our results suggest that linear disturbances can be reliably mapped using the FLM if the resolution of the input CHM is high relative to the size of the disturbance. In our analyses we observed that most legacy lines—commonly 10 m wide—were accurately mapped by the FLM using CHMs with resolution of 2 m or less, and low-impact lines—commonly less than 5-m wide—required CHMs twice as detailed for accurate mapping (see Figures 8 and 9). Therefore, we suggest as a rule-of-thumb that the input CHM should have a resolution of at least one fifth of the width of the forest lines being mapped. Wide linear disturbances such as those associated with roads, railways, pipelines, and power lines should be easily detectable with most commonly available aerial LiDAR datasets.

### 4.3. Line Attribution

Given the results presented for the MANOVA of line attributes generated by the proposed software tool, we believe that outputs of our workflow are imbued with sufficient information to distinguish between different line conditions, including treatment, size, and possibly other factors such as vegetation condition and wetness. The purpose of the line attribute analysis in this study was not to assess the accuracy of individual attributes in relation to the real measure of those factors in the field, but to simply assess if each factor and the union of several factors could be helpful in distinguishing

line types and ground conditions. For example, we did not measure the height and roughness of vegetation in the field to compare those measures to the predicted height and roughness according to our workflow. However, we expected lines with vegetation treatment to have taller and rougher vegetation than untreated lines, and that pattern is evident in the results we presented. Similarly, legacy lines were, predictably, much wider and had lower PAR than low-impact lines. Interestingly, treated lines were wider on average than their untreated counterparts of the same size class. We assume this effect is due to two factors: wider lines are more often targeted for treatment; and mounding may require more horizontal space than the usual size of low-impact lines.

Lines crossing wetlands were much wider than upland lines in general (160% wider on average). Lines treated with mounding, which is mostly a wetland type of treatment, also were much wider than other types of lines (127% wider on average). It is well known that lowland vegetation is much slower to regenerate than upland vegetation on linear disturbances [6,74]. Observations like these suggest that FLM outputs could help researchers understand the factors limiting forest restoration on seismic lines and provide a valuable assessment tool for resource managers doing restoration planning.

Since line attribution yielded predictable results that could be interpreted as functions of real-world factors (line size, treatment, environment) we consider it almost certain that these factors could be reliably predicted by a model using FLM-generated line attributes as predictor variables. More study is required in this context.

### 4.4. Limitations

As a semi-automated toolset, the FLM requires some operator interaction in order to work as intended. The first critical operator interaction is the creation and editing of seed points for creating LCPs and LCCs. If the vertices of the input lines are misplaced, local problems may occur. LCPs of line segments where the vegetation is denser on the line than in the adjacent forest, be it due to advanced line regeneration or to adjacent cut-blocks or lakes, can present local deviations outside the disturbances favoring areas with lower tree density. Another rare problem is LCPs can occasionally cut sharp corners by a few meters between sparse canopy or ignore small deviations on lines. Because of these limitations, we expect users to perform quality checks and refine inputs as needed (see flowchart of Figure 2) so that outputs are correct.

The second major operator interaction in the proposed workflow is the establishment of least-cost corridor thresholds (LCCTs) at each line. As a rule-of-thumb, low-impact seismic lines can be mapped with a fixed low LCCT, and larger classes of lines can be either mapped with fixed higher LCCTs or variable thresholds relative to adjacent forest density obtainable via a Zonal Threshold tool included in the FLM toolset. Setting up proper line thresholds requires familiarity with the workflow and/or trial-and-error iterations and therefore can be challenging and prone to local errors. Assuming the user can determine proper LCCTs via quality checks and iterative refinements (Figure 2) the proposed software tool is capable of mapping accurate and extensive linear disturbances. Ideally, future versions of the software would automate some of these steps to minimize human interaction.

Despite these current limitations, our assessment is that the time and effort required to produce accurate output products with the FLM is a tiny fraction of what would be required by fully manual operations.

### 4.5. Future Work

Line attributes obtained with the proposed software tool via spatial analysis of the center line, footprint polygon, and CHM could be used to generate classifiers and models to study the effect of treatments, predict regeneration trajectories and map line conditions across large application areas. Such efforts could contribute to wildlife studies, fragmentation analyses, biodiversity assessments, and restoration planning. Additional useful attributes could be incorporated to lines such as adjacent forest density, adjacent tree heights, line connectivity, and derived properties such as light incidence and relative line size to adjacent forest openings. The accuracy of line attributes relative to ground truth

could be assessed so that individual attributes could be used as direct estimations of real-world metrics. Network analysis of attributed lines could measure accessibility and thus help identify focal points which would benefit the most from CWD and mounding treatment, effectively diminishing predator and human use of the whole network. Line attributes, as well as classes, could be generalized to label large-scale regions such as townships, counties or National Topographic Series grids, enabling high-level management and planning. Such developments could greatly enhance our understanding of the impact of linear disturbances in forests and allow for more sophisticated strategies on vegetation management.

At the time of writing, the FLM relies on proprietary licenses, namely ArcGIS and its Spatial Analyst extension. By using these licenses, we shortened the development cycle of the FLM since all tools are integrated in the same set of ArcPy libraries–making them compatible amongst themselves–enabling us to focus on higher-level development. On the other hand, this limitation not only restricts the accessibility of the tool to the public but also increases processing times given the ArcPy library is notoriously slow to import. An important future step would be to translate the software tool to a fully open-source format, which would involve significant effort, likely making use of multiple open-source libraries and assuring their compatibility. Translating to user-friendly open source format is especially challenging when the target audience is not familiar with code distribution practices.

Some steps incorporated into the adopted workflow are relatively inefficient and could be replaced by more sophisticated methods in order to achieve faster run times, which could be useful in very large application areas (e.g., provincial or continental scale). For example, the cost distance tool in ArcPy is processing intensive and relatively inefficient to determine an LCP or an LCC because it considers all cells as potential path nodes. Cost distance is arguably the main bottleneck in the adopted workflow. This step could potentially be replaced by modified versions of Dijkstra or A-star path-finding algorithms [75]. Such a replacement would substantially lower the number of cell iterations and potentially remove intermediate steps such as the clipping of the cost-raster, reducing both processing and memory requirements.

We elected to compare the accuracy of FLM outputs in the boreal forest to those derived from a publicly available dataset generated by human photo-interpreters [69] since this type of linear-feature inventory is common to forests everywhere. However, we encourage other authors to test our findings in different forest types, and to compare the LCP approach we have adopted to other algorithms that appear in the literature.

## 5. Conclusions

We have presented a novel semi-automated workflow for extensively mapping high-resolution forest lines and linear footprint, as well as obtaining valuable line attributes for characterizing ground conditions. Our workflow utilizes the FLM toolset, which was designed by us to be accessible and open-source (though currently with ArcPy dependencies) and is intended to aid industry, scientists, and government personnel to better study the current state as well as future implications of linear disturbances in forests. Our accuracy tests showed that the developed tool reliably predicts line center and width (best case 11% and 22.7% mean deviation relative to width, respectively), being consistently more accurate than the best publicly available dataset in the area, produced by human photo-interpretation. Additionally, the FLM was able to generate CHM-derived attributes to individual linear features which can separate lines into different classes (legacy, low-impact, untreated, mounded, woody material), with statistical significance ($p < 0.001$). Our results suggest that this work may pave the way to enhanced vegetation-recovery predictions as well as management planning on a line-per-line basis in regional scale. We released the tool in a code-sharing platform in the hopes that collaborators and others may not only make use of the toolset, but also share their expertise suggesting new features, fixes, and improvements so that the whole community can benefit from an increasingly powerful toolset. We have learned that there are yet great advances to be made in linear-disturbance mapping in forests, that CHMs contain valuable information that is not used to its fullest potential in forest management, and that detailed line analysis in a continental scale may yet be possible.

**Supplementary Materials:** The source code for the Forest Line Mapper (FLM) toolset mentioned in this article is available online at https://github.com/appliedgrg/flm.

**Author Contributions:** Conceptualization, G.L.Q., M.M.R., G.J.M. and J.L.; methodology, G.L.Q. and M.M.R.; software, G.L.Q.; validation, G.L.Q. and M.M.R.; formal analysis, G.L.Q. and M.M.R.; investigation, G.L.Q.; resources, G.J.M. and J.L.; data curation, G.L.Q., J.L., and M.M.R.; writing—original draft preparation, G.L.Q., G.J.M. and M.M.R.; writing—review and editing, G.L.Q., G.J.M. and J.L.; visualization, G.L.Q.; supervision, G.J.M. and J.L.; project administration, G.J.M. and J.L.; funding acquisition, G.J.M. and J.L. All authors have read and agreed to the published version of the manuscript.

**Funding:** This research was supported by a Natural Sciences and Engineering Research Council of Canada Collaborative Research and Development Grant (CRDPJ 469943-14) in conjunction with Alberta-Pacific Forest Industries, Cenovus Energy, ConocoPhilips Canada, Canadian Natural Resources Ltd., and the Regional Industry Caribou Collaboration.

**Acknowledgments:** We would like to acknowledge the work and efforts of the people who contributed to past versions of the linear-disturbance mapper toolset. Jerome Cranston, formerly of Alberta Biodiversity Monitoring Institute (ABMI), came up with the original concept of line mapping using LCPs. Sarah Cole developed automated workflows with ArcGIS's Model Builder. Silvia Losada also contributed the work and created the first python prototypes of line mapping using LCPs.

**Conflicts of Interest:** The authors declare no conflict of interest.

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
