# Peer review of "The Forest Line Mapper: A Semi-Automated Tool for Mapping Linear Disturbances in Forests"

_remotesensing, doi:10.3390/rs12244176_

Round 1
Reviewer 1 Report
General comment
The paper is well written and describe form a technical point of view a new tool for mapping forest linear elements, as forest roads, trails, power lines etc. The authors indicated that these linear elements often represent the starting point of the forest damages and disturbances, thus they consider that mapping forest roads could help forest managers to implement forest management strategies to prevent or restore forest ecosystems. The paper is interesting and, in my opinion, could be published after minor revisions. Among others, the use of acronyms in the text is not consistently, thus you should use them after their introduction.
L35: “exploration and recovery” represent two of the most common forest management actions, thus I suggest to add the text “in other word forest management”.
L65-66: in the caption you describe very well the location of images reported in figure 1A, while there is not a geographic indication for pictures 1B and 1C. I suppose that the pictures are taken in the same area, could you add two points in figure 1A for indicating the geographic position of the landscape reported in the picture 1B ad 1C?
L83-85: could you provide a reference for this statement “In forests, for example, shadows and occlusions from surrounding trees commonly hinder the spectral, spatial, and textural patterns that most optical techniques rely on.”
L115: You introduce d the acronym CHM in the previous sentence, thus you should use the acronym hereafter. You made the same with other acronyms in the whole text. Please check and adapt in the whole text.
L169: In the caption and figure you mention the regional lines. Are they a limits (boundaries) of the region?
L198-200: This sentence “First, a binary canopy/no-canopy raster is created by applying a height threshold to the CHM, arbitrarily set at 1 meter.” in my opinion is not clear. what is a no-canopy raster? what is a height threshold, and why 1 meter. Please, could you explain it better?
L262: you already introduced the acronym CWD, you should use it after its introduction.
L273: It is not clear the number format. I found them in the result section (table 2), but here you have to clarify better the 3 numbers (maybe in this way: 4, 416 ha, 828 ha and 9,972 ha respectively). Please adapt it.
L314: “field surveys” which parameters were surveyed?
L358: “Version 10.7.1” Do you have any reasons for using 10.7.1 rather than 10.6 as in the previous elaboration (see LINE 302)? If yes please explain and maybe you can add some words in the limitation section. If no, I suggest to indicate the same ArcMAp version to foster the replication of the work.
L436: “..perimeter-area ratio (PAR)”. You already introduced it before. Please introduce the acronyms only the first time and then use them consistently.
Author Response
Thank you very much for the review. We will respond to individual comments below. In the following sections, (R) refers to reviewer comments; (A) represents the responses of the Authors.
(R) L35: “exploration and recovery” represent two of the most common forest management actions, thus I suggest to add the text “in other word forest management”.
(A) Sorry, but we do not equate “exploration and recovery” with “forest management”. As a result, we have elected to leave the passage as-is.
(R) L65-66: in the caption you describe very well the location of images reported in figure 1A, while there is not a geographic indication for pictures 1B and 1C. I suppose that the pictures are taken in the same area, could you add two points in figure 1A for indicating the geographic position of the landscape reported in the picture 1B ad 1C?
(A) Sorry, the photos are not from the same area, so we are not able to adopt the reviewer’s recommendation.
(R) L83-85: could you provide a reference for this statement “In forests, for example, shadows and occlusions from surrounding trees commonly hinder the spectral, spatial, and textural patterns that most optical techniques rely on.”
(A) We have added a reference for this statement.
(R) L115: You introduce d the acronym CHM in the previous sentence, thus you should use the acronym hereafter. You made the same with other acronyms in the whole text. Please check and adapt in the whole text.
(A) We have reviewed the document and made changes where appropriate. Thank you for catching this.
(R) L169: In the caption and figure you mention the regional lines. Are they a limits (boundaries) of the region?
(A) Elsewhere in the manuscript (L204) we specify that “regional-scale lines” are lines digitized at ~1:20,000. We have updated the figure caption to add this information. Hopefully this addresses the reviewer’s concern.
(R) L198-200: This sentence “First, a binary canopy/no-canopy raster is created by applying a height threshold to the CHM, arbitrarily set at 1 meter.” in my opinion is not clear. what is a no-canopy raster? what is a height threshold, and why 1 meter. Please, could you explain it better?
(A) The passage now reads “First, the CHM was classified into “canopy” and “no canopy” classes by applying a height threshold to the CHM, arbitrarily set at 1 meter.” Hopefully this is clearer.
(R) L262: you already introduced the acronym CWD, you should use it after its introduction.
(A) We have reviewed the document and made changes where appropriate. Thank you for catching this.
(R) L273: It is not clear the number format. I found them in the result section (table 2), but here you have to clarify better the 3 numbers (maybe in this way: 4, 416 ha, 828 ha and 9,972 ha respectively). Please adapt it.
(A) We have adapted it. Thanks for the suggestion.
(R) L314: “field surveys” which parameters were surveyed?
(A) The passage now reads “Each site was comprised of a 150-m stretch of seismic line, within which detailed field surveys measuring seismic-line centers and widths were made in the summer of 2017.” Hopefully this is clearer.
(R) L358: “Version 10.7.1” Do you have any reasons for using 10.7.1 rather than 10.6 as in the previous elaboration (see LINE 302)? If yes please explain and maybe you can add some words in the limitation section. If no, I suggest to indicate the same ArcMAp version to foster the replication of the work.
(A) The first passage was a typo. It is fixed now.
(R) L436: “..perimeter-area ratio (PAR)”. You already introduced it before. Please introduce the acronyms only the first time and then use them consistently.
(A) Fixed.
Reviewer 2 Report
Thank you for your compelling research on the development of a linear feature mapping tool for forest ecosystems. Mapping these features is, as you point out, very important for many reasons. I think your manuscript is good, and I wish you continued success with developing your tool set.
To improve your manuscript, I have left numerous comments embedded in the attached PDF. In general these contain minor questions or recommendations. My major concerns are:
- The comparison with existing, publicly available, photo-interpreted linear features is a keystone of this work. Therefore, I think it is worth describing the quality of these baseline data in more depth, including any information pertaining to variability in the accuracy of the photo-interpretated data, due to different imagery, different interpreters, or the existence of QA/QC practices. All of these factors are very influential in determining the quality of your reference dataset. Additionally, I think it is unclear what statistical test you used to compare the results to the public datasets. I see your accuracy/error calculations in equations 2 and 3, but how did you calculate statistical significance? This is critical to include.
- While I see the value of reporting the MAE as a percent of path width, I think it would be very useful to the reader to also report ranges of data units, i.e. meters of error. Since the physical units have direct meaning to users, this information is a useful complement.
- I don't fully understand Figure 9: why is there an upper limit on MAE % width? Couldn't the error exceed 100% of the path width? Currently this graph takes on a sigmoidal form, but without the constraint it appears to be a more familiar exponential. Clarifying this would be very useful.
Thank you, and good luck with your revisions.

Author Response
Thank you so much for your careful and insightful review. We have made numerous edits and additions to the manuscript on account of the comments embedded in the PDF. In addition, we will respond to your major comments below. In the following sections, (R) refers to reviewer comments; (A) represents the responses of the Authors.
(R) The comparison with existing, publicly available, photo-interpreted linear features is a keystone of this work. Therefore, I think it is worth describing the quality of these baseline data in more depth, including any information pertaining to variability in the accuracy of the photo-interpretated data, due to different imagery, different interpreters, or the existence of QA/QC practices. All of these factors are very influential in determining the quality of your reference dataset. Additionally, I think it is unclear what statistical test you used to compare the results to the public datasets. I see your accuracy/error calculations in equations 2 and 3, but how did you calculate statistical significance? This is critical to include.
(A) The reviewer is incorrect instating that the public dataset is our reference dataset; it is not. Our reference dataset are field-measured widths and centerlines described in 2.3.2 Reference Data. We compared our results to field data, not the public data set. We did the same thing with the public data: compared it to the reference (field) data. As a result, the thing the reviewer requests – a description of the quality of the public data – is exactly what we do in this manuscript. The accuracy of the public data set is reported in 3.2.1 Centerline spatial error and 3.2.2 Line footprint width.
What we do is compare the accuracy of the public dataset with the accuracy of the FLM outputs. We have added the following passage to 2.4.2 to specify the statistical test used to compare the MAE of the two datasets: “Differences in MAE between the public dataset and FLM outputs were tested for significance using a Z test.”
(R) While I see the value of reporting the MAE as a percent of path width, I think it would be very useful to the reader to also report ranges of data units, i.e. meters of error. Since the physical units have direct meaning to users, this information is a useful complement.
(A) We agree and have added second versions of Figures 8 and 9 to include absolute errors measured in m. Additional edits have been made to the text in cases where specific values have been mentioned. For example, the passage describing errors in line-footprint width now reads “The best-case scenario for legacy and low-impact seismic lines was 17.27% (1.19 m) and 27.41% (1.21 m) MAE respectively with 25-cm resolution inputs”
(R) I don't fully understand Figure 9: why is there an upper limit on MAE % width? Couldn't the error exceed 100% of the path width? Currently this graph takes on a sigmoidal form, but without the constraint it appears to be a more familiar exponential. Clarifying this would be very useful.
(A) Very good observation. The sigmoidal form of the original graph was an artifact of the confounding effect of line spacing. When CHMs became coarse enough, errors from adjacent lines began to bleed into each other. New versions of Figure 9 limit error measurements to 100% MAE (4m absolute MAE) to avoid this situation. The figure caption has also been supplemented with the following passage to explain: “Note that while width-error levels could exceed the values presented here, measurements were confounded by situations where line spacing began to approach 2x the resolution of the CHMs being tested. As a result, we limit our recording of errors here at 100% MAE (4m).“
Reviewer 3 Report
This paper introduced a new mapper tool: Forest line mapper (FLM), which is an open-source platform utilized for mapping and attributing the linear features from forest region. Besides the plane coordinates of linear features provided by previous datasets, FLM provides height information for linear features using LIDAR-derived height model, which can be applied in more analytical applications. It is an effective semi-automated method for the linear feature extraction, comparing to human photo interpreters.
My detailed comments are as follows:
1.In the part of ‘3.2. Accuracy assessments’, the whole study area is suggested to be included instead of just Kirby area.
2. In the part of “5. Conclusions”, the words for describing and the summarizing the work about FLM illustration should not occur. Just give out the findings of this paper and interpretations for the futural work.
3. The introduction need to be reconstructed.
Author Response
Thank you very much for the review. We will respond to individual comments below. In the following sections, (R) refers to reviewer comments; (A) represents the responses of the Authors.
(R) In the part of ‘3.2. Accuracy assessments’, the whole study area is suggested to be included instead of just Kirby area.
(A) Sorry but this is not possible. We only have reference data from the Kirby area.
(R) In the part of “5. Conclusions”, the words for describing and the summarizing the work about FLM illustration should not occur. Just give out the findings of this paper and interpretations for the futural work.
(A) There is nothing in the Conclusions describing an FLM illustration. We are not sure what the reviewer is referring to; sorry.
(R) The introduction need to be reconstructed.
(A) We disagree with this broad (and vague) assertion.
Reviewer 4 Report
- The paper introduced a newly developed tool for line feature extraction in forest environments. The tool may be useful for some areas of practitioners and researchers.
- The paper needs improvements to meet the criteria of academic papers.
- The table format is not adequate.
- The table titles are too long. Explanations should be added as footnotes.
- Figure 1C is unclear, and its order should be replaced with Figure 1B.
- The way of using "hypothesis" seems incorrect. If something is described as a hypothesis, it should be explained along with a mechanism behind the hypothesis. There should be some parts describing how the hypothesis is formulated. Please consider rephrasing it.
- Here are comments about the contents.
- Figure2: I don't know why it is called the "center" line. I feel "line feature" is a more common term. Also, it is not clear what "regional" lines mean and why they need to be called so. Additionally, I think "Draw/edit regional lines" means not Input (legend) but operation.
- L174 regional seed points -> In Figure 2, it is written as regional lines. Lines and points are technically not the same. Terms should be used consistently and precisely.
- L184: "Other users may wish to obtain footprint polygons, which provide a detailed outline of footprint boundaries and can be useful in studying line conditions and micro-habitats. " I don't understand well. Please use terms carefully such as line, outline, and boundary. Line and polygon are totally different as GIS terms.
- L207 Finally, the base e exponential -> the base exponential ?
- L213 "The vertices of regional-scale lines, manually digitized at a 1:20,000 scale, are used as seed points." -> I don't understand well. Please elaborate.
- Figure 3: how are the three options different?
- L272: The extent of their boundaries is 4 -> I don't understand what it means. Please elaborate. In addition, I think it should be described as four, not 4.
- Figure 4: I don't understand why the green color legend represents "boundary".
- Equation3: what is Wr?
- 4.4 human intervention: I don't think the term is appropriate. I think other terms such as preparation are more suitable. Please consider rephrasing it.
Author Response
Thank you very much for the review. We will respond to individual comments below. In the following sections, (R) refers to reviewer comments; (A) represents the responses of the Authors.
(R) The table titles are too long. Explanations should be added as footnotes.
(A) The tables do not have “titles”; they have captions. According to most style guides that we are familiar with, captions should be standalone – in other words, descriptive enough to not require the main text or footnotes. As a result, we have left our table captions as-is. We would be happy to re-visit this decision if directed so by the copy editor.
(R) Figure 1C is unclear, and its order should be replaced with Figure 1B.
(A) We’re sorry, we don’t understand this comment. Figure 1C shows something different than Figure 1B; replacing it would mean we lose the thing Figure 1C is trying to show. As a result, we are leaving the figure as-is.
(R) The way of using "hypothesis" seems incorrect. If something is described as a hypothesis, it should be explained along with a mechanism behind the hypothesis. There should be some parts describing how the hypothesis is formulated. Please consider rephrasing it.
(A) Good observation. We have augmented that paragraph with the following statement concerning our hypotheses: “To assist with our second and third objectives, we formulated a number of hypotheses based on expectations of how FLM outputs might compare to the best-available public dataset depicting linear features in our study area, plus a mechanistic understanding of how the resolution of input CHMs might influence FLM performance.”
(R) Figure2: I don't know why it is called the "center" line. I feel "line feature" is a more common term. Also, it is not clear what "regional" lines mean and why they need to be called so. Additionally, I think "Draw/edit regional lines" means not Input (legend) but operation.
(A) Regarding the term “centerline” – we are referring specifically to the line feature that runs down the center of the linear feature, and so feel that it is important to specify this. A “line feature” representing a road, seismic line, or other such entity does not necessarily need to run down the middle of it. As a result, we would prefer to keep the term.
Regarding the term “regional”, we agree that this is ambiguous. We have edited the text to specify our meaning here. For example, Line 173 now read “regional (i.e. lines digitized at ~1:20,000 scale) human-digitized linear feature databases”. We hope that this clarifies our meaning.
(R) L174 regional seed points -> In Figure 2, it is written as regional lines. Lines and points are technically not the same. Terms should be used consistently and precisely.
(A) Good catch. We have edited the caption of Figure 2 to refer to the points, not lines.
(R) L184: "Other users may wish to obtain footprint polygons, which provide a detailed outline of footprint boundaries and can be useful in studying line conditions and micro-habitats. " I don't understand well. Please use terms carefully such as line, outline, and boundary. Line and polygon are totally different as GIS terms.
(A) We agree – good catch again. We have replaced the term “line” in this passage with “linear feature”.
(R) L207 Finally, the base e exponential -> the base exponential ?
(A) Yes, we have edited the passage now to read “base exponential”.
(R) L213 "The vertices of regional-scale lines, manually digitized at a 1:20,000 scale, are used as seed points." -> I don't understand well. Please elaborate.
(A) We have updated the passage as follows:
“Using the CHM-derived cost raster described above, lines can be created at the center of linear disturbances by applying the LCP algorithm between two points, which we call “seed points”. Seed points can be created by manually digitizing linear features as lines at moderate scales (~1:20,000) and exporting the vertices. Public GIS datasets that represent linear features as lines can also be used as a source of seed points.”
We hope that this elaboration clarifies our meaning.
(R) Figure 3: how are the three options different?
We have updated the caption so as to better highlight the difference between the three options. The caption now reads:
“The Forest Line Mapper’s three segmentation options for defining the entities used for summarizing attributes: whole-lines, line-crossings (line intersections), and arbitrary. The whole-line option summarizes attributes across the entire line. The line-crossing option summarizes attributes between intersections. The arbitrary option summarizes attributes along arbitrary (in this case, 40m) segments.”
We hope this is clearer.
(R) L272: The extent of their boundaries is 4 -> I don't understand what it means. Please elaborate. In addition, I think it should be described as four, not 4.
(A) The passage was admittedly confusing. We have re-written it as the following:
“These areas are referred to as Kirby (416 ha in size), LiDEA I (828 ha) and LiDEA II (9,972 ha; n.b. LiDEA is an acronym for Cenovus Energy’s Linear Deactivation project)”
We hope that this is clearer now.
(R) Equation3: what is Wr?
(A) There was a typo in the equation. The term should have been Wir, and has been fixed now.
(R) 4.4 human intervention: I don't think the term is appropriate. I think other terms such as preparation are more suitable. Please consider rephrasing it.
(A) We have used the term “operator interaction” instead. We agree that the old term was vague.
Round 2
Reviewer 3 Report
- Author declared that the main target of this study is to analyze the performance of FLM toolset for mapping and attributing linear features in complex forest scenes. However, the “complex forest scenes” how to defined? It is very importance for the application of FLM toolset.
- “1.1 objectives” can be deleted in the part of introduction.
- The manuscript is difficult to read because of the poor English usage and awkward sentence structures. Author needs to significantly edit the manuscript to improve its readability.
- Table 1 should use the three-line table. Pleased revised it.
- “the Forest Line Mapper (FLM) toolset,” in the conclusion, the full name of FLM should be deleted due to the full name has been appeared in the manuscript.
- In this manuscript, amounts of the first person were used which is not suitable to the academic paper based on the objective description. Author should be revised.
- Does the method proposed is more optimal compare with other methods? I think that the comparison study is needed.
Author Response
Thank you for the additional comments. In the following sections, (R) refers to reviewer comments; (A) represents the responses of the Authors.
(R) Author declared that the main target of this study is to analyze the performance of FLM toolset for mapping and attributing linear features in complex forest scenes. However, the “complex forest scenes” how to defined? It is very importance for the application of FLM toolset.
(A) Thank you for the comment. We have augmented the passage in the objectives portion of the introduction to expand on what we mean by complex forest scenes. The passage (119-121) now reads as follows: “Our overarching goal was to evaluate the FLM toolset for mapping and attributing linear features in complex forest scenes with a variety of site types (upland/wetland) and stand characteristics”
(R) “1.1 objectives” can be deleted in the part of introduction.
(A) We have deleted the subheading label as requested.
(R) The manuscript is difficult to read because of the poor English usage and awkward sentence structures. Author needs to significantly edit the manuscript to improve its readability.
(A) We’re sorry, but we are unable to respond to this vague comment. We note that other reviewers described the manuscript as “well written”.
(R) Table 1 should use the three-line table. Pleased revised it.
(A) We have edited the format of the table. Thank you for the suggestion.
(R) “the Forest Line Mapper (FLM) toolset,” in the conclusion, the full name of FLM should be deleted due to the full name has been appeared in the manuscript.
(A) We have corrected this.
(R) In this manuscript, amounts of the first person were used which is not suitable to the academic paper based on the objective description. Author should be revised.
(A) We disagree. While there is some debate on whether science writing should use the active voice (“We have presented a novel semi-automated workflow…”) or passive voice (“A novel semi-automated workflow was presented…”) we prefer the former. Our view is that (i) the passive voice portrays a false sense of objectivity and (ii) the passive voice makes sentences opaque and difficult to read.
While MDPI does not specify a preference on active/passive voice in its style guide, many publications do. Nature journals, for example “prefer authors to write in the active voice ("we performed the experiment...") as experience has shown that readers find concepts and results to be conveyed more clearly if written directly” (see https://www.nature.com/nature-research/for-authors/write). This is the format we choose to write by, and recommend that the reviewer re-think their position.
(R) Does the method proposed is more optimal compare with other methods? I think that the comparison study is needed.
(A) We agree – that would be an interesting study. We have updated the “Future work” section (4.5) to include this good suggestion. The final paragraph in that section (677-681) now reads as follows:
“We elected to compare the accuracy of FLM outputs in the boreal forest to those derived from a publicly available dataset generated by heads-up digitizing [69] since this type of linear-feature inventory is common to forests everywhere. However, we encourage other authors to confirm our findings in different forest types, and to compare the LCP approach we have adopted to other algorithms that appear in the literature.”